# Comprehensive RNA-Seq Gene Co-Expression Analysis Reveals Consistent Molecular Pathways in Hepatocellular Carcinoma across Diverse Risk Factors

**DOI:** 10.3390/biology13100765

**Published:** 2024-09-26

**Authors:** Nicholas Dale D. Talubo, Po-Wei Tsai, Lemmuel L. Tayo

**Affiliations:** 1School of Chemical, Biological, and Materials Engineering and Sciences, Mapúa University, Manila 1002, Philippines; nddtalubo@mymail.mapua.edu.ph; 2School of Graduate Studies, Mapúa University, Manila 1002, Philippines; 3Department of Food Science, National Taiwan Ocean University, Keelung 202, Taiwan; powei@mail.ntou.edu.tw; 4Department of Biology, School of Health Sciences, Mapúa University, Makati 1203, Philippines

**Keywords:** hepatocellular carcinoma, histological grades, cancer etiology, module preservation, module enrichment, pathway analysis

## Abstract

**Simple Summary:**

The molecular heterogeneity of hepatocellular carcinoma (HCC) and its range of potential etiologies contribute to the complexities in treating this disease. Additionally, inter-sample molecular variability can mean the involvement of different prognostic genes or utilization of distinct molecular pathways in HCC development. This paper examines the genes and pathways most involved in different histological grades of HCC across various pre-cancer risk factors using publicly available bulk transcriptomics data and a systems biology approach. It identifies shared pathways among HCCs of varying grades and risk factors, as well as genes common to these pathways. Furthermore, this study highlights gene clusters preserved across risk factors, which may indicate shared targets for general treatment and gene clusters specific to viral or non-viral etiologies. Overall, this research reveals common and differing molecular pathways across risk factors and similarities in gene expression between histological grades. It provides a framework for understanding HCC development respective of risk factors and underscores the molecular pathways and genes involved.

**Abstract:**

Hepatocellular carcinoma (HCC) has the highest mortality rate and is the most frequent of liver cancers. The heterogeneity of HCC in its etiology and molecular expression increases the difficulty in identifying possible treatments. To elucidate the molecular mechanisms of HCC across grades, data from The Cancer Genome Atlas (TCGA) were used for gene co-expression analysis, categorizing each sample into its pre-existing risk factors. The R library BioNERO was used for preprocessing and gene co-expression network construction. For those modules most correlated with a grade, functional enrichments from different databases were then tested, which appeared to have relatively consistent patterns when grouped by G1/G2 and G3/G4. G1/G2 exhibited the involvement of pathways related to metabolism and the PI3K/Akt pathway, which regulates cell proliferation and related pathways, whereas G3/G4 showed the activation of cell adhesion genes and the p53 signaling pathway, which regulates apoptosis, cell cycle arrest, and similar processes. Module preservation analysis was then used with the no history dataset as the reference network, which found cell adhesion molecules and cell cycle genes to be preserved across all risk factors, suggesting they are imperative in the development of HCC regardless of potential etiology. Through hierarchical clustering, modules related to the cell cycle, cell adhesion, the immune system, and the ribosome were found to be consistently present across all risk factors, with distinct clusters linked to oxidative phosphorylation in viral HCC and pentose and glucuronate interconversions in non-viral HCC, underscoring their potential roles in cancer progression.

## 1. Introduction

Liver cancer is currently ranked as the sixth most common cancer as of the year 2024. Furthermore, it incurred the third most deaths among all cancers, and is the second leading cause of cancer-related mortality in men [1]. The high mortality rate and incidence of liver cancer has been consistent throughout the years, with hepatocellular carcinoma (HCC) consisting of 90% of all cases of liver cancer. As a result, massive efforts have been undertaken to understand the underlying mechanisms and pathophysiology of HCC.

HCC affects hepatocytes, which are known as the most common cell type of the liver. Hepatocytes, a type of parenchymal cell, are known to be involved in various metabolic processes, detoxification, and are crucial to maintaining liver homeostasis [2]. The role of the liver in the human body requires it to maintain persistent but regulated inflammation in order to facilitate its exposure to various dietary and pathogenic products and its resulting metabolic and tissue remodeling [3]. As such, the liver and its hepatocytes are known to have complex mechanisms and immunological structures necessary to regulate its microenvironment. However, such systems can fail, and dysregulation can lead to increased inflammation along with cell and organ damage. This increased inflammation eventually leads to cirrhosis, which is characterized by a larger ratio of fibrotic liver tissue, and is a known cause for HCC [4,5].

HCC exhibits great heterogeneity at the molecular level. Through transcriptomic analysis, subgroups of HCC have been defined, each with potential implications for biological pathways, clinical features, gene methylation, gene mutations, and chromosome loss of heterozygosity. Six subgroups were identified: Group 1 is characterized by patients with hepatitis B and with implications for developmental and imprinting genes like *IGF2*. Group 2 includes those enriched with AKT activation and mutations in the genes *AXIN1* and *PIK3CA*. In Group 3, mutations in *TP53* are observed. Group 4 is rare and includes those with mutations in *TCF1*. Finally, Groups 5 and 6 both include those exhibiting mutations in *CTNNB1* along with WNT activation. Further descriptions of each subgroup are available in Boyault et al. [6].

Inflammation and cirrhosis are involved in the development of hepatocellular carcinoma (HCC), but there are several other factors contributing to HCC that do not involve inflammation. Some of the most discussed etiologies include hepatitis B, hepatitis C, alcoholic liver disease, non-alcoholic liver steatohepatitis/non-alcoholic fatty liver disease (NASH/NAFLD), and aflatoxins [5]. These risk factors are not mutually exclusive, thus potentially adding further complications to the study of HCC. For infection-related etiologies, the oncoviruses hepatitis B (HBV) and hepatitis C (HCV) have been noted to drive HCC occurrence across various countries despite the availability of vaccines and treatments [7]. While termed as oncoviruses, HBV and HCV do not cause HCC directly; instead, the mechanisms necessary to complete their life cycles may eventually create the conditions ideal for hepatocarcinogenesis [8]. For HBV, hepatic cells are disrupted due to the virus’ ability to integrate within the host’s genome. Furthermore, HBV exhibits covalently closed circular DNA (cccDNA) that has proven to be able to remain stable in the host nucleus for a long time, thus facilitating chronic infection that is known to progress to liver diseases like cirrhosis [7]. Despite this, cases of non-cirrhotic HCC with an HBV etiology have been noted to exist, indicating its potential to dysregulate pathways important to carcinogenesis [9]. Unlike HBV, HCV is an RNA virus and does not integrate itself to the genome of infected hepatic cells. As expected, HCC with an HCV etiology is known to be preceded by liver fibrosis and cirrhosis, taking on the traditional liver disease pathway [7]. Chronic HCV further exacerbates the development of liver fibrosis and can increase the risk of HCC without treatment [10].

Nevertheless, the seroprevalance of HBV and HCV has been decreasing steadily in some high-risk areas to the point that alcohol-related or NASH/NAFLD HCCs are becoming the primary driver for HCC cases [1,11]. Alcohol-related HCC is known to be preceded by the established precancerous conditions of steatosis, steatohepatitis, and cirrhosis. However, specific to it is the formation of specific protein and DNA adducts due to the acetaldehyde, increased oxidative stress, and changes in the immune microenvironment and metabolism [12]. Interestingly, despite the non-alcoholic factor of NASH, it was noted to be histologically similar to alcoholic steatohepatitis [13]. The excessive accumulation of lipids characterizing NAFLD helps facilitate chronic inflammation due to endoplasmic reticulum stress and is known to lead to hepatocarcinogenesis [14].

The complex etiology of HCC possibly contributes to the notorious tumor heterogeneity of HCC. HCC has been known to exhibit inter-patient heterogeneity (IPH), intratumor heterogeneity (ITH), spatiotemporal heterogeneity, and etiological heterogeneity [15,16]. The heterogeneity of HCC has been implicated in its limited treatment options and clinical intractability [17]. Evidence of morphological intratumor heterogeneity has also been found where more than two histological grades were found in more than half the samples [18]. The increased tumor heterogeneity and the potential complex etiology of the samples must be taken into account in further studies about HCC.

In this paper, the gene expression of HCC tumors was examined across its reported pre-cancer risk factors, which point to potential etiology, and histological grades. Unlike other studies, both the viral and non-viral risk factors were considered. Gene clusters found by weighted gene co-expression analysis are correlated with the histological grades of various samples grouped by their risk factor. The enriched functions of the genes potentially point to pathways and biological processes involved in the progression of HCC. Furthermore, it can also identify clusters unique per group potentially identifying specific targets for personalized treatments of HCC cases.

## 2. Materials and Methods

### 2.1. Retrieval and Preprocessing of Data

The dataset used in this study was retrieved from The Cancer Genome Atlas (TCGA) database. The raw expression data of the cohort TCGA-LIHC were retrieved along with the corresponding clinical metadata with the R library TCGAbiolinks. Since the exact nature of the pre-existing conditions of the samples from TCGA were reported for some of the samples, the metadata was used to group the samples by pre-existing condition. The groups included hepatitis C, hepatitis B, alcohol consumption with no other condition, NAFLD, and no history of risk factors. The sample IDs were also filtered out for the presence of normal samples, recurrent and metastatic tumors. The final dataset consisted of a gene matrix with raw count values and the sample IDs, which has grade and condition information. Overall, all the available HCC samples in the TCGA-LIHC cohort in 2024 were retrieved, in which only 303 viable samples were used in this study.

As suggested in WGCNA workflows, genes were filtered out to remove noise. The Bioconductor package BioNERO was used for the processing and construction of the co-expression networks, with WGCNA as its algorithm [19]. Each group counts and metadata were placed in a corresponding SummarizedExperiment object and were filtered for missing data using the *removena* function of BioNERO. The library DESeq2 was then used for variance stabilizing transformation specifying covariates as gender and tumor grades. Non-expressed genes or those who had a median of lower than five were removed. The transformed counts were then subjected to mean filtering and the top 8000 most varying genes were taken per condition. Afterwards outliers in the sample were removed using ZKfiltering [20]. After the successful conversion and transformation of both matrices, the result was taken into the WGCNA pipeline.

### 2.2. Gene Co-Expression Network Construction

To begin, a range of powers were set to be explored, particularly one to twenty with increments of two. Using the pickSoftThreshold function of the WGCNA package, the scale-free topology fit index and the mean connectivity of each dataset were calculated. Powers above 0.80 scale-free topology fit index were considered. Furthermore, the value of the mean connectivity for each was ensured to be below 100. The number of bins in connectivity histogram was set to 10 and network type signed was used as the input parameters.

The workflow followed the standard procedure in constructing weighted gene co-expression networks for each of the objects and was done through the exp2gcn function of BioNERO. The connection strength between the genes was calculated with the adjacency matrix with a set network type of signed. Furthermore, a module merging threshold of 0.4 was set. For all groups, the power ten successfully fulfilled the required conditions and was then chosen. The resulting adjacency matrix was then used to find the interconnected genes through signed type of topological overlap matrices (TOM). The TOM of each object was used to find their dissimilarity matrix (1-TOM). Through the dissimilarity matrix, the hierarchical cluster of genes were found with the UPGMA algorithm [21].

### 2.3. Gene Co-Expression Network Analysis and Hub Gene Identification

After the construction of the network, the dendrogram and module colors of all groups were plotted. Both the unmerged and merged modules were shown in the plot. The count of genes per module was determined to provide a clearer visualization of the size of each module. The module–trait correlation heatmap was plotted and the most correlated modules per tumor grade were noted per condition excluding the module grey. The hub genes of the relevant modules were taken using the get_hubs_gcn function of BioNERO which considers genes whose degree is at the top 10% and has a Module Membership (MM) > 0.8. The drug-target data was extracted from Drugbank 5.1.12 to determine the presence of the targets in various grades.

Besides examining the modules related to historical grades, the preservation of the modules per dataset was also calculated. The reference dataset was the no history dataset due to its potential to have no viral and non-viral condition-specific modules that may skew the results. This means that the datasets hepatitis C, hepatitis B, alcohol consumption with no other condition, and NAFLD were set as the test set. Similar to other studies, a permutation of 100 was done and the resulting values were retrieved [22]. Furthermore, the modules per risk factors were compared to each other using hierarchical clustering. A presence matrix was created from the list of genes per module. Using the presence matrix, the distance matrix was calculated using the dist function with method binary. The optimal number of clusters was determined using the silhouette method, in which the resultant value was used for the hclust function with the parameter complete for the method. Each clusters of modules was enriched with clusterProfiler.

### 2.4. Over Representation Analysis (ORA)

To find the potential functions of the modules of interest, ORA was considered best for the dataset and its known use for functional enrichment [23]. The R library clusterProfiler was used to perform ORA on the chosen modules. The enrichGO and enrichKEGG were used to functionally enrich the clusters in the databases Gene Ontology and Kyoto Encyclopedia of Genes and Genomes (KEGG) database, respectively [24]. By dividing the cluster by their risk factors and grading information, the compareCluster function allowed the use of a faceted dotplot visualization. Furthermore, information about each hub gene were queried with the MyGene.info API [25].

## 3. Results

The completion of the preprocessing step per dataset outputs a consistent data for all five datasets. First, it was confirmed that all datasets only consisted of the top 8000 most variable genes. Furthermore, the clustering and filtering of the samples left 96 samples for the hepatitis B dataset, 46 samples for hepatitis C, 71 samples for no history, 18 samples for NAFLD, and 62 samples for alcohol consumption. All samples were well within the required number for WGCNA analysis.

Appendix A summarizes the results of the scale free topology model fit of each of the datasets. This helps further reduce noise for the module identification after the network construction. As observed, the soft threshold power of nine was above 0.80 for each of the datasets. Furthermore, its mean connectivity is well below 100. As such, this value was chosen, and all datasets were confirmed for analysis.

The WGCNA algorithm uses the dynamic tree cut algorithm for module detection. Appendix A summarizes the results for the module detection of all datasets. For context, the unmerged and merged modules were shown for all the datasets. By examining each of the dendrogram, it can be observed that for all conditions no genes made it past a height of 0.50. As the height axis in dendrograms serves to depict the dissimilarity between its leaves, the lack of any group of genes making it past 0.50 indicates that there is a lower correlation between the expression of each gene within the samples of each condition. This will also be reflected by the number of genes per modules per condition.

Figure 1 is a grouped plot of histograms that summarizes the number of genes per modules. The bars of the histogram corresponds to the assigned color designation of the module. Due to the preprocessing steps of this study, the overall number of genes should be the same for all conditions. However, the modules detected by the algorithm have no limit other than the set minimum module size and module merging height, which combines the similar modules to each other. From the results above, it appears that NAFLD-HCC has the highest number of modules at eleven. It is then followed by the HCC-AC and HCC-HepB datasets at 10 modules. Lastly, conditions HCC-HepC and HCC-NH were found to be the lowest at 7 modules. Observing Appendix A, it can be found that those with less and higher hanging leaves had more modules present compared to the others.

Nevertheless, by applying a consistent merging threshold, a reasonable number of modules for analysis can be retrieved. The modules in each dataset were then correlated with their respective phenotypes, particularly the gender of the samples and their histological grade. This approach allows for the examination of the most relevant modules per histological grade by selecting the ones most correlated, regardless of *p*-value. The results of the correlations were generally weak. As observed in Figure 2, the modules that achieved a moderate correlation (greater than or equal to 0.4) per grade were the brown and darkorange2 modules from the alcohol consumption dataset and the black module from the no history dataset. Weaker but significant correlations were found in other datasets, such as the blue module in the no history dataset, and the hepatitis B and hepatitis C datasets. Despite the generally weak correlations, the most correlated modules per grade across risk factors were still examined, as they may have biological relevance that is obscured by the heterogeneous nature of HCC.

Each gene present in the modules most correlated with each histological grade was retrieved and functionally enriched. Figure 3 presents a dotplot summary of the top enriched terms per ontology and grade. For Biological Processes, catabolic processes for small molecules, organic acids, and carboxylic acids are enriched in Grade I HCC across the datasets of alcohol consumption, hepatitis B, hepatitis C, and NAFLD. Following that, the terms chemotaxis and cytoplasmic translation were enriched in Grade II HCC. For Grade III HCC, terms related to nuclear division and chromosome segregation were found to be enriched in the datasets, with overlap observed in Grade IV HCC of the HCC-HepB dataset. In contrast, for the HCC-HepC dataset, Grade IV genes were enriched in extracellular structure organization, extracellular matrix organization, and external encapsulating structure organization.

For Cellular Components, Grade I HCC was enriched with terms such as blood microparticle, peroxisome, and microbody across some datasets, with the exception of HCC-HepB, where the term mitochondrial matrix appeared. The gene products of Grade II HCC were predominantly found in the collagen-containing extracellular matrix and the ribosomal unit. For Grade III HCC, genes were consistently observed in chromosomal regions, with overlap noted in Grade IV HCC.

The enriched molecular functions varied for Grade I HCC, including terms such as monooxygenase activity, oxidoreductase activity acting on CH-OH group donors, and vitamin binding. For Grade II HCC, the results were more consistent, with most genes belonging to the term extracellular matrix structural constituent. Lastly, the same overlap observed in the enrichment of Biological Processes and Cellular Components was seen in Grade III and Grade IV HCC, with terms such as DNA-binding transcription activator activity, RNA polymerase II-specific DNA-binding transcription activator activity, and single-stranded DNA helicase activity being enriched.

Similar patterns from the GO results can be observed with Figure 4. For Grade I HCC, the most correlated genes appear to be involved in metabolic pathways. In particular, KEGG pathways involving lipid metabolism, metabolism of cofactors and vitamins, xenobiotics biodegradation and metabolism, amino acid metabolism, and carbon metabolism were found. Furthermore, across all etiologies, the KEGG term “Complement and coagulation cascades” consistently appears for Grade I HCC. All of this indicates the involvement of metabolism in early HCC, either due to metabolic programming or as a side effect of the late-stage liver disease that characterizes the precancer environment.

For Grade II HCC, the enriched KEGG terms can be grouped into signaling molecules and interaction, development and regeneration, immune system, and signal transduction. The wide scope of these terms indicates an increasing complexity in cancer and the potential dysregulation of more pro- and anti-tumor pathways. With WGCNA, the genes most correlated with Grade II HCC describe a gene cluster most associated with the grade, suggesting the involvement of genes from ECM-receptor interaction, cell adhesion molecules, and other similar terms, indicating increased invasion as is expected from the increased aggressiveness of Grade II HCC. However, for HCC-HepC, the terms enriched for Grade II HCC differed, with oxidative phosphorylation and thermogenesis being prominent.

Some consistency was also observed with the results of Grade III and Grade IV HCC. The enriched terms could be grouped by their involvement in cell growth and death and replication and repair. The pathways involved are notably classic in cancer, with genes involved in cellular senescence and the p53 signaling pathway. For Grade IV HCC, the terms are either involved in similar pathways to Grade III or to the terms related to the ECM and cell adhesion. The choice of the group of genes to enrich for each grade of HCC and the results does not indicate that these pathways only activate at this point, but rather that they have increased activity compared to other gene clusters. The increased aggressiveness of the cancer and its high differentiation from normal cells could explain the relevance of these pathways and their corresponding genes for Grade III and Grade IV HCC.

The top 10 hub genes of the selected modules from the gene co-expression network were retrieved for further analysis. These hub genes were ranked by their kWithin values, which, if high, indicate that a gene is highly interconnected within its module. Furthermore, it can also be expected that these genes will have a function related to the terms enriched in Figure 3 and Figure 4. The shared genes across the hub genes per dataset were retrieved. In the case of Grade I HCC, it appears that genes *PCK2*, *CYP8B1*, *ACSM2A*, and *HAAO* were present in more than one dataset. Following that, for Grade II HCC, genes *STING1* and *LCP2* were identified. Lastly, for Grade III HCC, genes of the Kinesin family like *KIF23*, *KIFC1*, *KIF11*, and *KIF23* were found across the top 10 hub genes. Other shared genes were also found for Grade III HCC, including *GINS1* and *TOP2A*. Due to the small number of samples of available Grade IV HCC tumors, only the HCC-HepB and HCC-HepC had phenotypic information on this histological grade. Nevertheless, hub genes for these were identified, and they appear to either share the Kinesin family hub genes of Grade III HCC or the ECM genes of Grade II HCC. As mentioned in the methodology, hub genes are those genes whose degree is at the top 10% and have a Module Membership (MM) > 0.8. However, for the sake of brevity, only the top 10 were placed in Table 1 and their shared genes were identified. The complete list of hub genes per grade and condition is available in the Appendix A in Appendix A.

The genes most related to each histological grade under different conditions were compared with the known interactions of various HCC drugs. As observed, the interactions of the selected drugs are predominantly associated with Grade 1 and Grade 2 HCC. However, common drugs like Sorafenib and Regorafenib appear to interact with genes across all grades. On the other hand, lesser known but approved drugs such as Atezolizumab and Bevacizumab primarily interact with genes related to a subset of histological grades. Although these interactions are visualized in Figure 5, their precise significance remains unclear. Thus, while Sorafenib may interact with genes across all histological grades, its primary target might be more closely related to a specific grade. For example, one of Sorafenib’s targets is the gene *FLT4*. This gene was found to be involved with histological grades I, II, and IV in at least one or more of the risk factors considered. A summary of the targets of each drug is available in Appendix A.

To further elucidate the similarities and differences between each risk factor, the modules were clustered and biologically enriched. Hierarchical clustering enabled the identification of groups of modules with similar functions across the datasets. A cluster count of 13 was determined to yield the highest silhouette value (Appendix A), indicating that modules are more closely matched within their own clusters than with those in other clusters. As observed in Figure 6, most modules diverge at a distance greater than 0.7, reflecting the dissimilarities among the various modules. Each cluster is assumed to be biologically meaningful, and the pathways most active in these clusters can be observed through enrichment analysis. Additionally, Figure 6 may account for biological deviations, as only clusters 3, 4, 5, 6, and 7 present a complete set of the risk factors considered. Interestingly, genes that are not part of any modules were found to cluster primarily in cluster 7. In contrast, clusters 2 and 10 lack representation from “no history” and “alcohol consumption” risk factors, respectively. The other clusters seem to be combinations of different risk factors. However, clusters 1 and 11 is notable for them specific to non-viral and viral risk factors, respectively. Table 2 summarizes the top enrichment pathways for various clusters, each associated with a specific biological category.

As observed in Table 2, Cluster 1 is enriched in carbohydrate metabolism, particularly in pentose and glucuronate interconversions, with an adjusted *p*-value of 0.000601. Cluster 2 is linked to xenobiotics biodegradation and metabolism, specifically the metabolism of xenobiotics by cytochrome P450, showing a highly significant enrichment with a *p*-value of 2.14 × 10^−8^. Cluster 3 is related to the immune system, with enrichment in complement and coagulation cascades (*p*-value 1.48 × 10^−20^). Cluster 4, involved in translation, is significantly associated with the ribosome pathway (*p*-value 2.06 × 10^−44^). Cluster 5 highlights cell growth and death, particularly the cell cycle (*p*-value 2.35 × 10^−24^), while Cluster 6 focuses on signaling molecules and interaction, with cell adhesion molecules being the top pathway (*p*-value 5.21 × 10^−24^). Cluster 7 is associated with amino acid metabolism, particularly arginine and proline metabolism (*p*-value 0.01008). Cluster 8, although unnamed, is enriched in pathways related to the cytoskeleton in muscle cells (*p*-value 8.83 × 10^−14^). Cluster 9 is involved in membrane transport, with ABC transporters being the top pathway (*p*-value 0.002854). Cluster 10 pertains to cellular community in eukaryotes, focusing on tight junctions (*p*-value 0.010539). Cluster 11 is linked to energy metabolism, particularly oxidative phosphorylation (*p*-value 0.019103). Cluster 12 also deals with amino acid metabolism, specifically the degradation of valine, leucine, and isoleucine (*p*-value 0.000382). Finally, Cluster 13 is associated with the nervous system, with serotonergic synapse showing the highest enrichment, although this is less significant with a *p*-value of 0.054565.

In order to compare and identify the modules shared by HCC of various pre-existing risk factors, module preservation analysis was conducted with the no history dataset as the reference network. The choice of the reference network is primarily determined by the apparent lack of a driving biological condition for the tumors in the no history group. As such, it can reasonably be asserted that the biological modules formed by the no history dataset will not have exclusive modules arising due to a pre-existing condition, allowing the potential identification of shared target genes. Appendix A summarizes the module preservation results. As observed, most modules show high preservation, with some of their Z _summary_ values reaching as high as 30. This affirms the preservation of the blue, black, cyan, dark magenta, and dark grey modules. The brown module has a preservation that appears consistently Z < 10. To prioritize the most preserved modules, only those with Z > 20 across all datasets were retrieved.

The modules selected for further analysis were the black and cyan modules. The pathways enriched by the gene members of these modules, along with the gene co-expression network, are shown in Figure 7. For most of the datasets, it can be observed from Appendix A that the cyan module has a generally higher preservation score than the black module. Through functional enrichment, it was observed that the cyan module corresponded to pathways potentially involved in cellular interaction with its environment. KEGG terms like cell adhesion molecules and cytokine-cytokine receptor interaction were enriched and are known to be involved in cancer invasion or metastasis [26]. On the other hand, the black module appears to be related to genes involved in the regulation of the cell cycle and apoptosis. The preservation of such genes is expected as they form the core mechanism of cancer. They also align with the complete risk factor membership of cluster 5 and cluster 6.

## 4. Discussion

In a clinical setting, histological grades and pathological stages are important predictors of the biological or pathological behavior of cancers [27]. The ability to classify cancers into groups provides medical professionals with the opportunity to understand the clinical outcomes of cancer and offer appropriate treatments. While histological grading and pathological staging have provided high prognostic value in terms of patient survival, their exact value at a molecular level is rarely shown in the most accessible resources. For hepatocellular carcinoma, studies on biomarkers for each histological grade of the cancer are available. However, it appears that the identification of the most significant pathways per grade, along with the potential examination of gene expression differences in HCC from differing sources, has not yet been undertaken [28,29].

It should be noted that the identification of the most correlated gene clusters to each histological grade and their functional enrichment does not imply that these pathways are only present or relevant at a particular grade. The complexity of cancer progression means that various dysregulated pathways and the genes involved can play multiple roles in different tumor types or environments. For example, in this study, *PCK2* was found to be a hub gene in Grade I HCC. However, experimental methods have shown its involvement not just in metabolic-related pathways but also in pathways involving immune regulation, proliferation, and metastasis of HCC [30]. This indicates that the gene has multiple roles across various histological grades. This is further supported by the observed enrichment pattern wherein metabolic-related, immune, and ECM-related pathways appear to switch between Grade I and Grade II HCC across the datasets. A similar pattern can be observed between Grade III HCC and Grade IV HCC. The similarity in the expression of G1/G2 and G3/G4 observed in the hierarchical clustering results (Figure 2) suggests that while certain pathways can be quantified as most related to a histological grade, the activity and significance of these pathways in the overall biology of the tumor may not decrease as the histological grade increases. This observation will guide further discussion of the results in this paper. Lastly, Figure 8 summarizes the discussion by examining each pathway present in each grade and the overlapping shared hub genes of relevant pathway as seen in Appendix A.

### 4.1. Grade I and Grade II HCC

Histological Grade I and Grade II HCC should together present as a less aggressive tumor for afflicted patients. As mentioned earlier, these two grades appear to have metabolic-related, immune, and ECM-related pathways, which is supported by the current understanding of carcinogenesis and early cancers [31,32]. However, the etiology of HCC potentially confers the involvement of metabolism even in precancerous conditions. For example, nonalcoholic fatty liver disease (NAFLD) is pathologically known for metabolic dysregulation, to the point where it was recently renamed as metabolic dysfunction-associated fatty liver disease (MAFLD) and is known to involve the PPAR signaling pathway along with HCC [33,34]. This means that these pathways may potentially already be dysregulated before the point of hepatocarcinogenesis occurred.

Nevertheless, the enriched pathways in Figure 4 are related to the metabolic reprogramming that hepatocytes undergo in hepatocellular carcinoma (HCC). This reprogramming can be divided into three major areas: carbon metabolism, lipid metabolism, and amino acid metabolism [35]. Reprogramming carbohydrate metabolism is crucial to support rapid tumor growth compared to normal cells. In HCC, tumor cells switch to glycolysis due to its faster rate compared to oxidative phosphorylation (OXPHOS) [35,36]. Genes involved in glycolysis were found in modules of various networks. Interestingly, the hub genes *PCK1* and *PCK2* were found to be implicated not only in glycolysis but also in the PPAR signaling pathway and the PI3K/Akt pathway. All three pathways appear to be involved in the metabolic reprogramming of HCC, with the PI3K/Akt pathway promoting glycolysis while inhibiting gluconeogenesis, the PPAR signaling pathway modulating the PI3K/Akt pathway, and glycolysis playing a central role [37,38]. The involvement of these pathways was also observed in other cancers like gliomas with the exception of the PPAR signaling pathway [39].

Furthermore, the metabolism of propionate has been shown to promote fatty acid oxidation (FAO) in liver cells, counteracting the effects of oxidative stress [40]. EHHADH is known to promote FAO, and suppressed FAO has been shown to increase the aggressiveness of HCC cells. Additionally, FAO provides energy for anti-inflammatory macrophages [41].

Expectedly, there is an overall functional overlap between Grade 2 HCC and Grade I. For example, changes in the calcium signaling pathway in hepatocytes can worsen fat buildup and potentially promote the progression of HCC. This pathway is involved in one of the potential etiologies of HCC, specifically MAFLD [33,42]. Another overlap is observed with the PI3K/Akt pathway, which is involved in both the regulation of metabolism and immune responses. This pathway specifically involves the *IL7R* gene, which is known to be related to tumor immunity and acts as a signal receiver that helps the immune system recognize and attack HCC cells [30,43].

The central role of the PI3K/Akt pathway is further highlighted by the fact that genes within this pathway are shared and activated with the ECM, promoting cancer cell survival and proliferation. Conversely, the PI3K/Akt pathway also influences ECM remodeling, facilitating tumor invasion and metastasis [44].

### 4.2. Grade II and Grade III HCC

For higher-grade HCC, the enriched terms start to involve classical cancer-related genes of cell proliferation, cellular senescence, and metastasis. The p53 signaling pathway appeared enriched, and genes *CCNB1*, *CCNB2*, *CDK1*, and *CHEK1* were identified as overlapping hub genes, indicating the aberrant involvement of cell cycle arrest and apoptosis in response to cellular stresses like DNA damage [45]. Differential gene expression analysis found *CDK1* to be differentially expressed and related to histological differentiation and serum CEA. The same study also identified *CCNB1* and *CCNB2* as DEGs and pathologically related to cirrhosis and serum albumin [46]. As these genes are involved in cell cycle regulation and G2/M transition, an increase in their overall expression indicates the increasing aggressiveness of the G3/G4 group.

In the most aggressive classification, tumor samples classified as Grade IV HCC appear to be present only for HCC-HepB and HCC-HepC. Furthermore, the two modules most correlated seem to have differing sets of genes and corresponding pathways. Nevertheless, there appears to be a biological precedent with the correlated modules. In particular, the involvement of the cell adhesion and focal adhesion KEGG terms and the overlapping hub gene *ITGA9* solidify the existence of a highly aggressive tumor. *ITGA9* has been found to be an important mediator of tumor metastasis for HCC and other human tumors, serving not only as a potential prognostic factor but also as a therapeutic target specifically for suppressing metastasis in HCC [47]. Its importance, along with its overlap with genes in the PI3K/Akt signaling and adhesion pathways, indicates that Grade IV HCC may utilize similar pathways to Grade II HCC to initiate its metastatic activity, coupled with the continuous dysregulation of the cell cycle observed greatly in Grade III HCC.

### 4.3. Module Preservation and Clustering

Through hierarchical clustering, the overall similarities of the identified co-expression modules per etiology was examined. Interestingly, across all risk factors modules related to the cell cycle, cell adhesion, the immune system, and the ribosome was found. This supports the findings above and points to the primary shared mechanisms of HCC. Furthermore, the clustering yielded a cluster specific only to viral and non-viral etiologies. First, both hepatitis B and hepatitis C appeared to have a co-expression module for oxidative phosphorylation. The significance of these genes could be related to the impairment of oxidative phosphorylation and reduced ATP production brought on by increased lactic acidosis and oxidative stress brought on by HBV or HCV carcinogenesis [48,49]. For the non-viral specific cluster, it was found to be enriched in Pentose and glucuronate interconversions. This pathway is known to be induced by ethanol and involved in alcoholic fatty liver disease [50]. Furthermore, the same pathway is downregulated in the pathogenesis of NAFLD [51]. Overall, the results of the hierarchical clustering provides insights to the difference between various pre-cancer risk factors that was not captured by the module–trait correlation.

When examined across tumors with different historical risk factors, the modules with the highest preservation are those related to the cell cycle and adhesion, which consequently are the KEGG terms most enriched in the G3/G4 group and clusters 5 and 6. By applying module preservation, this study attempts to determine a shared druggable group of genes should the etiology be unclear. Overall, the results align with the involvement of the p53 signaling pathway across tumor types [45]. The importance of the dysregulation of cell adhesion genes in different tumors is consistent with the results found throughout literature [47]. These pathways may serve as potential targets for HCC cases regardless of etiology.

The clusters and the specific preserved modules show some overlap with the molecular study of TCGA on HCC. In that study, HCC was clustered into three distinct subtypes using multi-omic data, including DNA methylation, RNA, miRNA, and proteomic expression. Somatic alterations in signaling pathways, such as the cell cycle and PI3K pathways, were observed, potentially explaining the significance of these pathways in our study [52]. Additionally, some hub genes identified in this study, like *CCNB1*, were also implicated, as it was found to be underrepresented in a specific cluster of samples. While TCGA’s HCC study provides a multi-omic perspective on the cancer, it did not examine gene coexpression. Other studies have attempted to address this gap by applying WGCNA to TCGA data. For example, one study used WGCNA on normal and tumor tissue, identifying a module enriched in metabolic and cell cycle pathways, with *CCNB1* emerging as a significant hub gene—similar to our findings [53]. Similar pathways were also found in an HBV-HCC specific study [54]. However, the same study how they relate to etiology. Our findings further confirm the activation of critical pathways, like cell cycle and cell adhesion. Unlike previous works, however, our study attempts to provide deeper insight into the roles of these pathways across different phenotypes (etiology and histological grade) through the incorporation of module–trait correlation, hierarchical clustering, and module preservation. The resulting specific clusters of modules found for viral and non-viral risk factors potentially point to the differences in the resulting molecular expression of HCC depending on the pre-existing risk factors.

## 5. Conclusions

Given the aggressive nature of HCC and its poor clinical prognosis, there is an increasing need to identify effective treatments for hepatocellular carcinoma (HCC) with diverse pathologies. However, the heterogeneous nature of this cancer poses a challenge in finding an approach that can treat HCC consistently across and within patients. This study aimed to examine on a transcriptome level, the activity of the most variable modular co-expressed genes across different risk factors and histological grades of HCC. Pathways and hub genes were identified that are involved in the regulation of metabolism, cell signaling, the immune system, cell cycle, and cell adhesion. Furthermore, the enriched pathways were found to be consistent across the datasets and adjacent histological grades. The modules found were subjected to hierarchical clustering in order to determine significant clusters per risk factor. It was found that modules related to the cell cycle, cell adhesion, the immune system, and the ribosome are present across all risk factors. Furthermore, clusters of modules specific to viral or non-viral pre-cancer risk factors were observed to be enriched in oxidative phosphorylation and pentose and glucuronate interconversions, respectively. Modules enriched in cell cycle and cell adhesion genes were found preserved indicating the central role they play. The identification of hub genes in the co-expression network provided a means to examine the potential interplay between the dysregulated pathways of HCC. Overall, the results of this study elucidate the role of different pathways in the progression of HCC and point to potential target pathways and genes for therapeutic action. Furthermore, co-expression modules unique to viral and non-viral HCC were found, potentially elucidating their relevance in the progression of the cancer.

## Figures and Tables

**Figure 1 biology-13-00765-f001:**
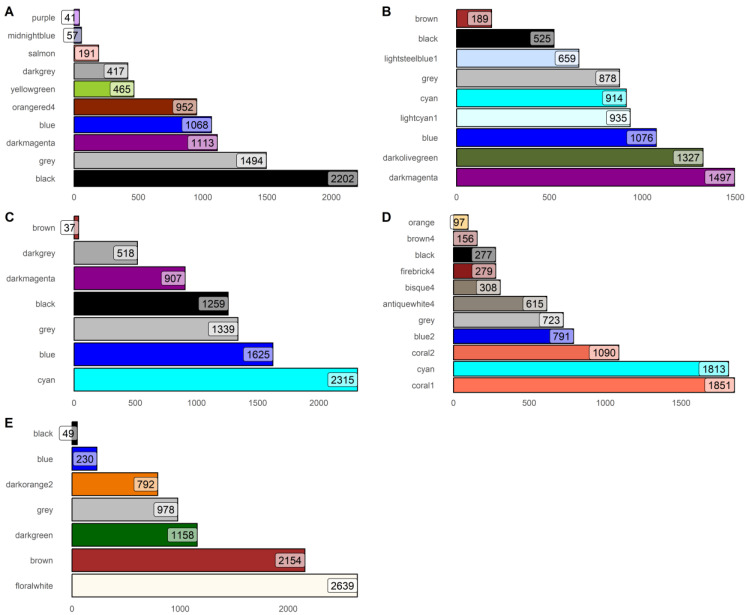
Number of modules per dataset with (**A**) alcohol consumption (**B**) hepatitis B, (**C**) hepatitis C, (**D**) NAFLD, (**E**) no history.

**Figure 2 biology-13-00765-f002:**
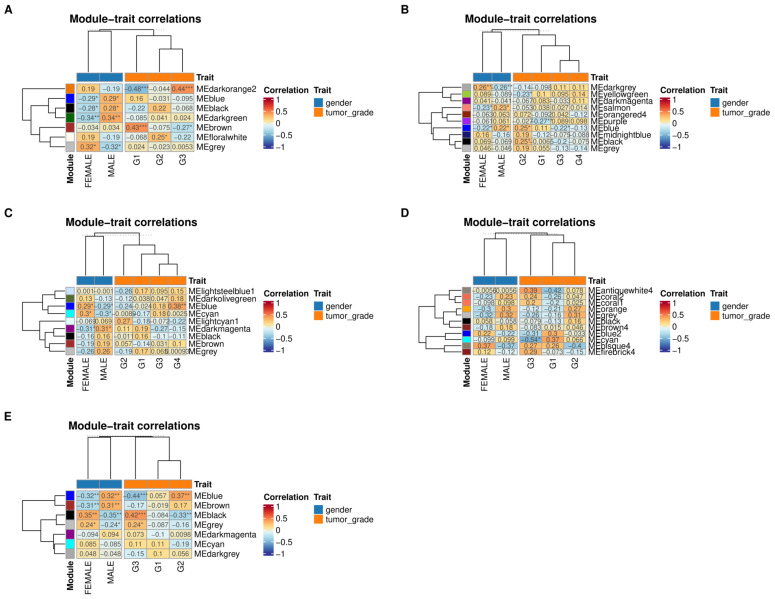
Module–trait correlation results per dataset with (**A**) alcohol consumption (**B**) hepatitis B, (**C**) hepatitis C, (**D**) NAFLD, (**E**) no history. Asterisk depicts significance starting at less than 0.05 at one asterisk to less than 0.001 at three.

**Figure 3 biology-13-00765-f003:**
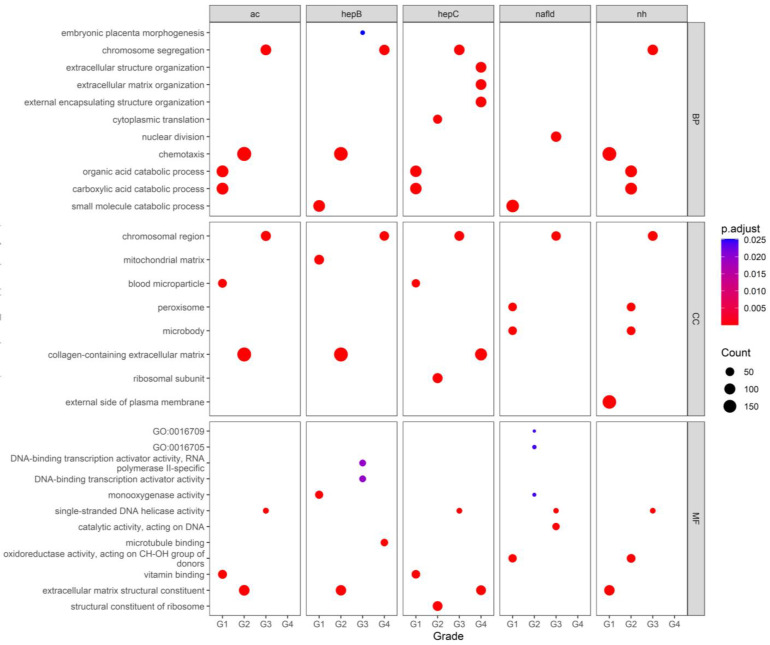
Faceted dotplot of the Gene Ontology enrichment results from clusterProfiler. The results for Biological Process (BP), Cellular Component (CC), and Molecular Function (MF) significantly enriched in selected gene clusters are visible per datasets.

**Figure 4 biology-13-00765-f004:**
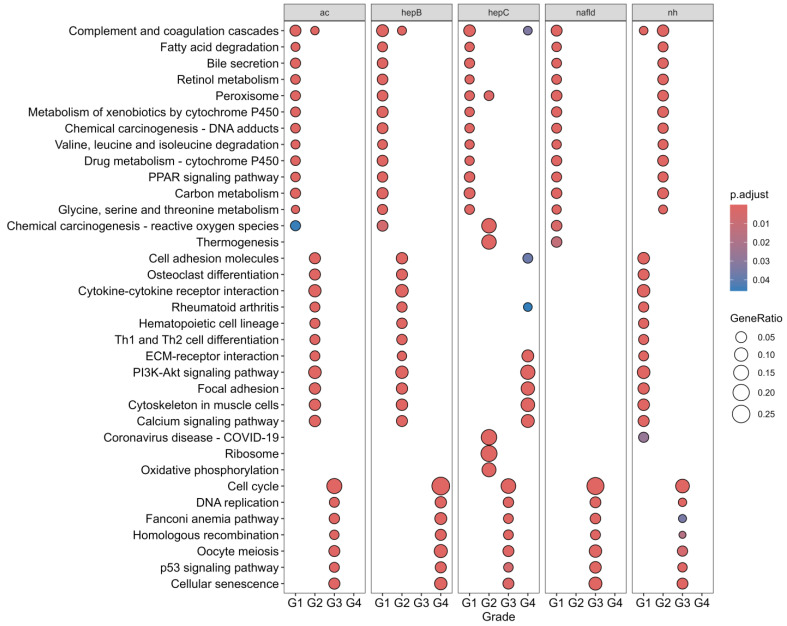
Faceted dotplot of the KEGG enrichment results from clusterProfiler. The results for KEGG significantly enriched in selected gene clusters are visible per datasets.

**Figure 5 biology-13-00765-f005:**
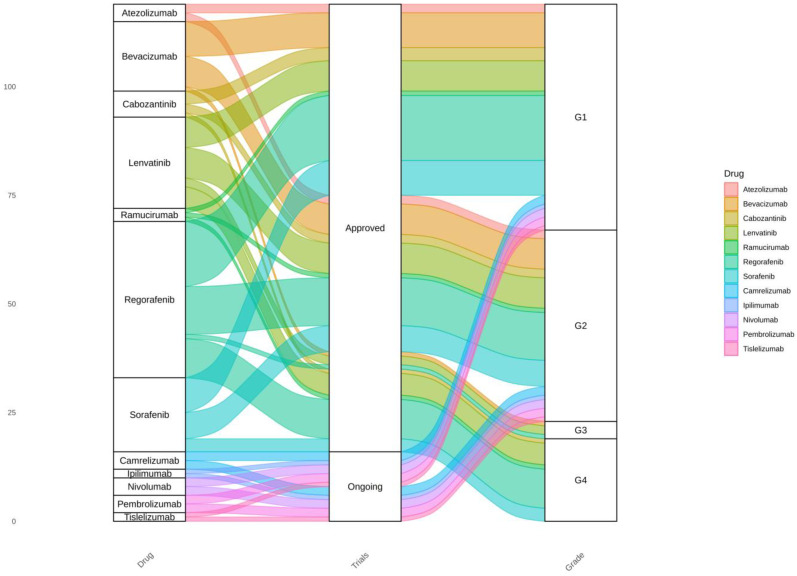
Alluvial diagram depicting the relationship between approved and in-trial drugs for HCC and the membership of their known gene interactions across different histological grades.

**Figure 6 biology-13-00765-f006:**
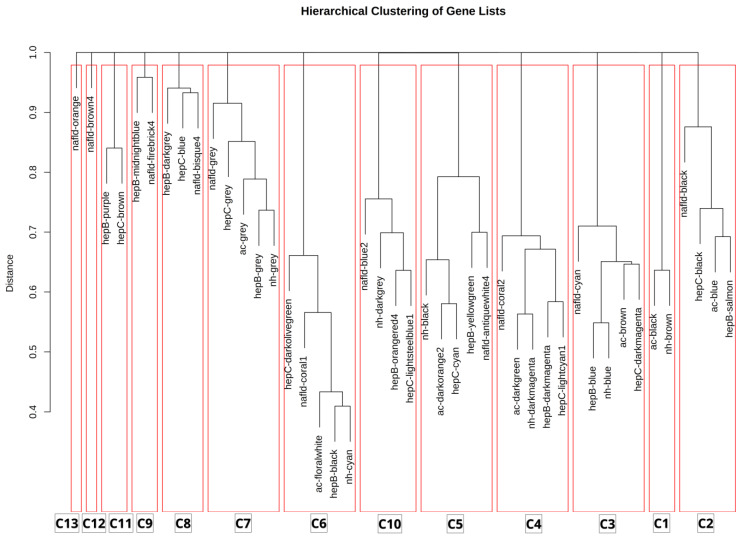
Hierarchical clustering of co-expression modules of all the risk factors considered.

**Figure 7 biology-13-00765-f007:**
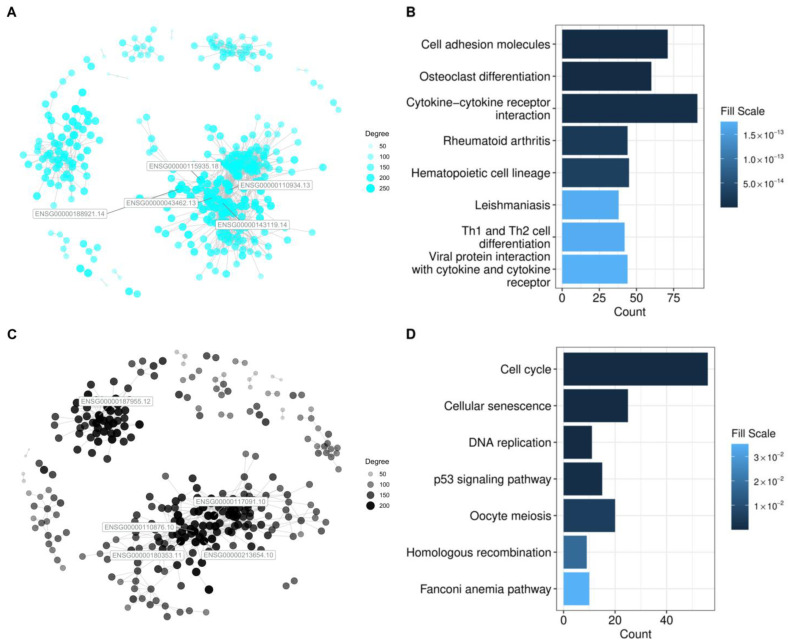
Gene co-expression network of the black and cyan module, placed in (**A**) and (**C**), along with corresponding enrichment barplot (**B**) and (**D**).

**Figure 8 biology-13-00765-f008:**
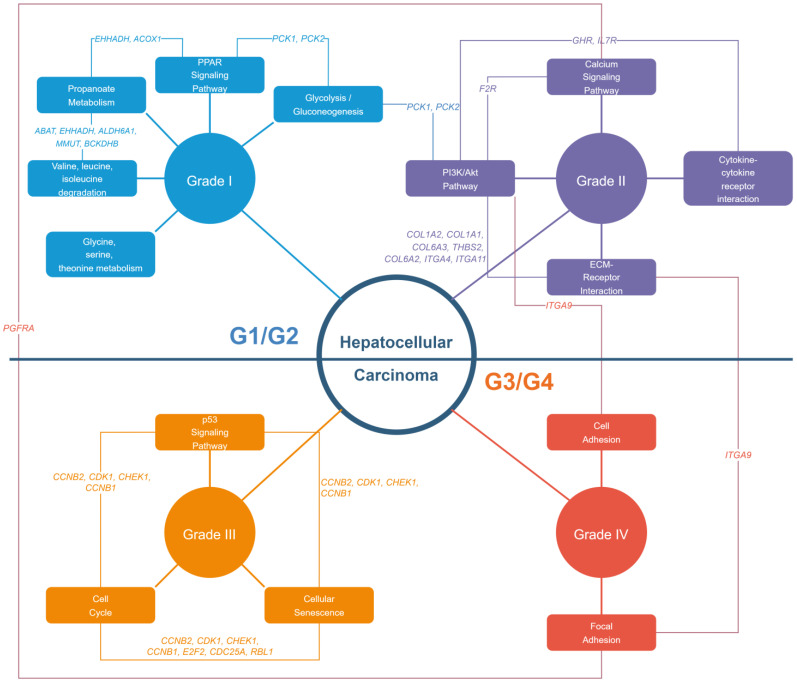
Gene–concept network based on module–trait correlation results, grouped by HCC grades. Each KEGG term is connected to a histological grade and to other pathways through a hub gene shared by both.

**Table 1 biology-13-00765-t001:** Top 10 hub genes for different histological grades and risk factors.

Histological Grades	Alcohol Consumption	Hepatitis B	Hepatitis C	Non-Alcoholic Fatty Liver Disease (NAFLD)	No History
Grade I	*ABAT*, *SCP2*, *PFKFB1*, *ACSM2A*, *HSD17B6*, *PCK2*, *SLC10A1*, *EHHADH*, *ALDH6A1*, *CYP8B1*	*GLYAT*, *SLC27A5*, *PCK2*, *CYP8B1*, *SEC14L2*, *ASPDH*, *ACSM2A*, *HAGH*, *HAAO*, *LDHD*	*TFR2*, *CYB5A*, *SERPINC1*, *RBP4*, *PCK2*, *HPN*, *HAAO*, *RIDA*, *SDC1*, *APOF*	*KNG1*, *AASS*, *PLG*, *TF*, *IVD*, *CYP4V2*, *C6*, *TLCD4*, *ACAA1*, *BCKDHB*	*LCP2*, *HACD4*, *WIPF1*, *BIN2*, *CD53*, *HCLS1*, *IKZF1*, *EVI2B*, *NCKAP1L*, *TSHZ3*
Grade II	*STING1*, *HACD4*, *COL3A1*, *RAB31*, *LCP2*, *HEPH*, *GPR132*, *ADAMTS2*, *ZEB2*, *KCTD12*	*STING1*, *HCLS1*, *SELPLG*, *CD48*, *GPSM3*, *COL14A1*, *LCP2*, *GLIPR2*, *ANTXR1*, *COTL1*	*FAU*, *ELOB*, *GADD45GIP1*, *ATP5F1D*, *BLOC1S1*, *NDUFA3*, *CHCHD5*, *SERF2*, *ROMO1*, *EDF1*	*C3orf18*, *TMEM139*, *CYP2C18*, *TNFSF10*, *NA*, *ZNF69*	*EHHADH*, *SEC14L2*, *GLYATL1*, *SLC27A5*, *CYP8B1*, *DMGDH*, *PCK2*, *ACSM2A*, *GLYAT*, *FMO4*
Grade III	*CCNB2*, *HJURP*, *TPX2*, *KIFC1*, *KIF23*, *KIF11*, *KIF4A*, *NEK2*, *TTK*, *MELK*, *CCNB1*	*WNK2*, *MAPK13*, *EPCAM*	*KIF18B*, *KIF23*, *TOP2A*, *KIFC1*, *CKAP2L*, *NCAPH*, *MCM10*, *GINS1*, *CDCA8*, *TICRR*	*GINS1*, *TOP2A*, *CLSPN*, *KIF11*, *NUF2*, *KIF2C*, *CDK1*, *ZWINT*, *KIF23*, *SGO1*	*KIF18B*, *CDK1*, *KIF23*, *CENPA*, *TOP2A*, *GTSE1*, *TICRR*, *MYBL2*, *KIF18A*, *TROAP*
Grade IV	*-*	*KIF18B*, *KIF4A*, *KIF23*, *TPX2*, *CKAP2L*, *GINS1*, *TOP2A*, *BUB1*, *HJURP*, *MELK*	*COL14A1*, *ANTXR1*, *PODN*, *COL3A1*, *KIRREL1*, *LAMA2*, *SSC5D*, *EPHA3*, *AEBP1*, *ISLR*	*-*	*-*

**Table 2 biology-13-00765-t002:** Top-most biologically enriched pathways per module.

Clusters	Category	Top Enrichment	*p*-Value Adjusted
Cluster 1	Carbohydrate metabolism	Pentose and glucuronate interconversions	0.000601
Cluster 2	Xenobiotics biodegradation and metabolism	Metabolism of xenobiotics by cytochrome P450	2.14 × 10^−8^
Cluster 3	Immune system	Complement and coagulation cascades	1.48 × 10^−20^
Cluster 4	Translation	Ribosome	2.06 × 10^−44^
Cluster 5	Cell growth and death	Cell cycle	2.35 × 10^−24^
Cluster 6	Signaling molecules and interaction	Cell adhesion molecules	5.21 × 10^−24^
Cluster 7 ^1^	Amino acid metabolism	Arginine and proline metabolism	0.01008
Cluster 8		Cytoskeleton in muscle cells	8.83 × 10^−14^
Cluster 9	Membrane transport	ABC transporters	0.002854
Cluster 10	Cellular community— eukaryotes	Tight junction	0.010539
Cluster 11	Energy metabolism	Oxidative phosphorylation	0.019103
Cluster 12	Amino acid metabolism	Valine, leucine, and isoleucine degradation	0.000382
Cluster 13	Nervous system	Serotonergic synapse	0.054565

^1^ Cluster 7 is composed of grey modules and is therefore a list of genes that are not co-expressed.

## Data Availability

All data used in this study were retrieved from the The Cancer Genome Atlas Program (TCGA) through the use of the R package TCGAbiolinks and the Genomic Data Commons (GDC).

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
