# Peer review of "Comprehensive RNA-Seq Gene Co-Expression Analysis Reveals Consistent Molecular Pathways in Hepatocellular Carcinoma across Diverse Risk Factors"

_biology, 2024, doi:10.3390/biology13100765_

Round 1

Reviewer 1 Report

Comments and Suggestions for Authors

The reviewed article is devoted to the study of molecular heterogeneity of hepatocellular carcinoma based on the analysis of a large array of transcriptome data from The Cancer Genome Atlas (TCGA) using a set of adequate systems biology approaches. The main goal of this study is to identify key genes and signaling pathways that may be involved in the development and progression of hepatocellular carcinoma at different histological stages. Of particular importance to this work is the fact that a separate analysis of transcriptomes of different groups of tumors was carried out, differing in potential precancerous risk factors, such as infection with hepatitis B virus, infection with hepatitis C virus, the presence of alcoholic liver disease or non-alcoholic steatohepatitis/non-alcoholic liver disease (NASH/NAFLD). As a result of the preliminary analysis, samples of the initial experimental data were selected, the detailed analysis of which using bioinformatics methods allowed to identify genes whose expression is changed during tumor progression. Sets of genes characteristic of four histological stages (G1, G2, G3 and G4) of HCC were obtained in five groups corresponding to four risk factors and their absence. Finally, an enrichment analysis of gene clusters in the gene databases of the Gene Ontology (GO) and Kyoto Encyclopedia of Genes and Genomes (KEGG) databases allowed to bringing together G1/G2 and G3/G4 by the involvement of individual molecular pathways in the development of HCC. These are the metabolic regulation pathways and the PI3K/Akt pathway in the case of G1/G2 and the cell adhesion gene activation pathways and the p53 pathway in the case of G3/G4. The article presents the results of an original study of both theoretical and practical interest. The results are reliable. The results of the work can be used to select potential markers and therapeutic targets both common to different etiological groups of HCC and specific to individual groups with different risk factors.

Reviewer 2 Report

Comments and Suggestions for Authors

Overall, This paper looks good, I would like to say it could be accepted by Biology after minor revision.

While the paper mentions Weighted Gene Co-expression Network Analysis (WGCNA), it would be beneficial to provide more detailed information on how the analysis was conducted. This includes the criteria for selecting gene modules, thresholds for correlation, and any normalization steps taken.

Do we have more background data to prove your paper, and how about the other literature?

Reviewer 3 Report

Comments and Suggestions for Authors

This manuscript presents standard bioinformatics analyses of transcriptomic data from HCC patients obtained from the TCGA portal. The manuscript reads reasonably well, but there is a question of novelty and overlap with published work, see below. The authors split the available data in several groups of pre-existing conditions, e.g. infection by HBV / HCV or NASH/NAFLD. In the end, they observed that almost identical molecular pathways were active in the various groups. Although this may be worth reporting, it is sort of a negative result.

Points to be addressed in a revision:

(1) The authors do not Which ones of their findings overlap with the main
TCGA paper about the HCC data: https://doi.org/10.1016/j.cell.2017.05.046

(2) There is plenty of related bioinformatics research in the literature. Very little of this is cited in the manuscript.

When I simply googled for "TCGA HCC WGCNA", this brought up, among others, the following works:

https://www.frontiersin.org/journals/genetics/articles/10.3389/fgene.2020.00153/full
https://www.medsci.org/v20p0870.htm
https://www.frontiersin.org/journals/genetics/articles/10.3389/fgene.2022.1017551/full
https://infectagentscancer.biomedcentral.com/articles/10.1186/s13027-021-00357-4
https://jgo.amegroups.org/article/view/63832/html
https://www.mdpi.com/2073-4425/9/2/92
https://www.spandidos-publications.com/10.3892/ol.2023.14044
https://journals.lww.com/md-journal/fulltext/2022/10280/exploring_novel_independent_prognostic_biomarkers.65.aspx
https://www.jcancer.org/v12p1884.htm

Most of these studies seem to have used transcriptomic data of HCC samples from TCGA and processed it with the WGCNA package, as was done in the current work.
So one would expect that some of the findings of the present study should overlap with what has been reported before. This should discussed.

Some of the previous works focused on sub-groups of HCC patients, e.g. the ones with pre-existing HBV infection. The others analyzed all data. Given that the study under consideration found consistency between the molecular pathways
active for various risk condition, I would expect a reasonable overlap with the previous analyses.

The authors need to mention some of these previous works and discuss what distinguishes their studies from the other ones.

(3) The authors could move several figures to the supplement that are of less interest to the readership of this journal. I suggest to move Fig. 1, Fig. 2 and Fig. 7.

(4) To make this work a bit more interesting, the authors could discuss whether target proteins of available HCC drugs are expressed at different grades (G1 - G4) or not. E.g. https://www.cancer.gov/types/liver/what-is-liver-cancer/treatment
lists as treatments for advanced liver cancer:
 bevacizumab, cabozantinib, lenvatinib, ramucirumab, regorafenib, sorafenib

(5) line 137: why do you remove outliers after applying DESeq2? Usually, outliers should be removed first.

(6) Many of the correlations reported between lines 227 - 232 are very moderate to low. I would not mention correlations below 0.4.

Comments on the Quality of English Language

line 29 into their -> into its

line 30: insert "samples" after "For those"

line 207: insert "genes per" after "Number of"

line 382: noun seems to be missing after "known"

line 399: were -> was
